# Identification of Long Non-Coding RNA-Associated Competing Endogenous RNA Network in the Differentiation of Chicken Preadipocytes

**DOI:** 10.3390/genes10100795

**Published:** 2019-10-12

**Authors:** Lan Chen, Tao Zhang, Shanshan Zhang, Jinxing Huang, Genxi Zhang, Kaizhou Xie, Jinyu Wang, Haiqing Wu, Guojun Dai

**Affiliations:** 1College of Animal Science and Technology, Yangzhou University, Yangzhou 225009, China; chenlan9326@163.com (L.C.); zss18352764956@163.com (S.Z.); hjinxing1225@163.com (J.H.); gxzhang@yzu.edu.cn (G.Z.); kzxie@yzu.edu.cn (K.X.); jywang@yzu.edu.cn (J.W.); hqwu@yzu.edu.cn (H.W.); daigj@yzu.edu.cn (G.D.); 2Joint International Research Laboratory of Agriculture and Agri-Product Safety, Ministry of Education, Yangzhou University, Yangzhou 225009, China

**Keywords:** long noncoding RNA, preadipocytes differentiation, competing endogenous RNA network, chicken

## Abstract

Emerging evidence indicates that long noncoding RNAs (lncRNAs) play important roles in the regulation of cell differentiation by acting as competing endogenous RNA (ceRNA). However, the regulatory mechanisms of lncRNA and the lncRNA-associated ceRNA network involved in adipogenic differentiation of chicken preadipocytes remain elusive. Here, we first constructed the chicken preadipocyte in vitro induction model. Then, we identified differentially expressed lncRNAs (DELs), miRNAs (DEMis), and mRNAs (DEMs) between differentiated and undifferentiated preadipocytes. Furthermore, we constructed the lncRNA associated ceRNA network by gene expression correlation analysis and target prediction of DELs, DEMis, and DEMs. Finally, we determined twelve candidate lncRNA-miRNA-mRNA interactions from the lncRNA associated ceRNA network. Eight out of the twelve interactions were validated by RT-qPCR, indicating their potential role in the regulation of chicken preadipocytes differentiation. Among the eight interactions, TCONS_00026544-gga-miR-128-1-5p-RASD1, TCONS_00055280-gga-miR-135a-5p-JAM3, TCONS_00055280-gga-miR-135a-5p-GPR133, TCONS_00055280-gga-miR-135a-5p-CLDN1, and TCONS_00055280-gga-miR-135a-5p-TMEM123 may promote adipogenic differentiation of chicken preadipocytes while TCONS_00057272-gga-miR-146a-3p-FOXO6, TCONS_00057242-gga-miR-6615-3p-FOXO6, and TCONS_00057242-gga-miR-6615-3p-ENSGALT00000043224 have the opposite effects. Our results not only provide novel insights into ceRNA roles of lncRNAs in chicken preadipocytes differentiation and but also contribute to a better understanding of chicken fat deposition.

## 1. Introduction

In the past few decades, the growth rate of broilers has been significantly improved due to continuous selection [1]. However, the high growth rate is accompanied by excessive fat deposition [2]. Excessive fat deposition (mainly abdominal fat) could influence the laying rate, carcass yield, feed conversion ratio, hatching rate, and fertility rate. Therefore, more attention is paid to reducing abdominal fat, which has become a major breeding goal in broilers [3]. In addition, study showed that the adipose tissue of chicken has a relative resistance to insulin, and chicken could be used as an animal model to study human type 2 diabetes [4]. Thus, understanding the biological basis of the development of chicken adipose tissue can benefit the development of poultry industry and also provide new idea for human biomedical research. In mammals, the molecular and cellular mechanisms of adipose tissue development have been well studied. However, the mechanisms underlying chicken fat tissue development are complex and remain elusive.

Long non-coding RNAs (lncRNAs) are a class of non-coding RNA greater than 200 nucleotides in length. MicroRNAs (miRNAs) are a class of non-coding single-stranded RNA of approximately 22 nucleotides in length encoded by endogenous genes. miRNAs play essential roles in various life activities, including metabolism [5], development, and disease [6]. Chicken abdominal fat tissue development is a result of the increase in cell number (hyperplasia) and the increase in cell size (hypertrophy or preadipocytes differentiation). Recent studies have shown that LncRNAs and miRNAs participate in regulating preadipocytes differentiation in mammals [7,8]. Their roles in the differentiation of chicken preadipocytes are still poorly understood.

The competitive endogenous RNA (ceRNA) hypothesis proposes that transcripts can regulate each other at the post-transcription level by competing for shared miRNAs [9]. lncRNA can act as a miRNA sponge via competing endogenous RNA (ceRNA) activity [10], thereby regulating the gene expression of miRNA [11]. Understanding the RNA interaction will lead to significant insights into gene regulatory network in the differentiation of chicken preadipocytes. The role of lncRNA as ceRNA in adipocytes differentiation of mammal is gradually becoming clear. Liu et al. [12] found that lncRNA Gm15290 sponges miR-27b to promote PPARγ-induced fat deposition in mice. Li et al. [13,14] confirmed that lncRNA GAS5 negatively regulates the adipogenic differentiation of MSCs and 3T3-L1 cells by modulating miR-18a and miR-21a-5p as ceRNAs. Li et al. [15] reported that lncRNA ADNCR suppresses adipogenic differentiation by targeting miR-204 in bovine. However, the regulatory mechanisms of lncRNA as ceRNA and the lncRNA-associated ceRNA network involved in adipogenic differentiation of chicken preadipocytes remain elusive.

To identify lncRNAs and lncRNA-associated ceRNA network involved in differentiation of chicken preadipocytes, we systematically analyzed the differentially expressed lncRNAs, miRNAs, and mRNAs using RNA-seq and miRNA-seq. We then constructed the lncRNA-associated ceRNA network following the ceRNA hypothesis. Crucial ceRNA interactions were identified based on the interacted miRNAs function and validated using quantitative reverse transcription polymerase chain reaction (RT-qPCR). Our results not only provide novel insights into ceRNA roles of lncRNAs in chicken preadipocytes differentiation and but also contribute to a better understanding of chicken fat deposition.

## 2. Materials and Methods

### 2.1. Ethics Statement

This experiment was performed in accordance with Chinese guidelines for animal welfare, and the animal protocol was approved by the animal welfare committee of Yangzhou University (permit number SYXK (Su) 2012-0029).

### 2.2. Cell Culture and Induction

Three Haiyang Yellow chickens were anaesthetized with sodium pentobarbital and killed at two weeks old. Primary preadipocytes were isolated from the abdominal fat tissue following methods described by Shang et al. [16]. Complete medium (90% DMEM/F12 medium,10% fetal bovine serum, 100 units/mL penicillin, and 100 μg/mL streptomycin) was used to culture chicken preadipocytes in a cell incubator with 5% CO_2_ at 37 °C. when the cell reached 90% confluence, the differentiation medium (supplemented with oleic acid (400 μmol/L)) was used to induce the differentiation of chicken preadipocytes. Undifferentiated (0 days) and differentiated (6 days) cells were collected. Each interval was carried out for three biological replicates. The differentiation of chicken preadipocytes was determined using RT-qPCR and Oil red O staining. After induction for six days, the cells were fixed in 10% formalin for 30 min at room temperature. Then the cells were washed using PBS buffer for three times and stained using Oil red O working solution for eight minutes. Finally, the cells were washed using PBS buffer and observed using a microscope. The expression level of adipocyte marker genes PPARG and FABP4 in undifferentiated and differentiated preadipocytes was determined using RT-qPCR. The HiScript II Q RT SuperMix for qPCR (+gDNA wiper) kit (Vazyme, Nanjing, China) was used to synthesize the first-strand cDNA following the manufacturer’s protocol. The AceQ qPCR SYBR Green Master Mix kit (Vazyme, Nanjing, China) was used to perform RT-qPCR following the manufacturer’s protocol.

### 2.3. RNA Extraction and Library Construction

Trizol reagent (Tiangen, Beijing, China) was used to extract total RNA from undifferentiated and differentiated chicken preadipocytes (Add 1 mL of Trizol per 2.5 × 10^6^ cells). The NEBNext Poly(A) mRNA Magnetic Isolation Module kit (New England Biolabs, Ipswich, MA, USA) and NEBNext Small RNA Library Prep Set for Illumina (Multiplex Compatible) kit (New England Biolabs, Ipswich, MA, USA) were used to construct the lncRNA libraries and miRNA libraries following the manufacturer’s protocol, respectively. Six lncRNA libraries (three for each group) and six miRNA libraries (three for each group) were constructed. The 12 constructed libraries were sequenced using the Illumina HiSeqTM 4000 sequencing platform (Illumina Inc., San Diego, CA, USA).

### 2.4. LncRNA Identification

The raw data were filtered by removing low-quality reads (*Q*-value ≤ 20), reads containing adapters, reads that are all A bases, and reads containing unknown nucleotides ratio greater than 10%. The clean data were aligned with rRNA sequences of chicken using Bowtie2 to remove reads mapped to rRNA [17]. Then the high-quality clean data were mapped to the Gallus gallus reference genome using Tophat2 (version 2.0.3.12) to identify novel transcripts [18] and known lncRNAs. New lncRNAs were predicted based on the novel transcripts using CPC [19], CNCI [20], and SwissProt database [21]. The arguments and the thresholds used to identify lncRNAs were specified in Appendix A.

### 2.5. miRNA Identification

We filter the raw data by removing the low-quality reads (*Q*-value ≤ 20), reads containing adapters, reads shorter than 18 nt, and reads with polyA. Then reads were aligned with small RNAs in the GenBank [22] and Rfam [23] database using blastall software (version 2.2.25). The clean data was obtained by removing reads that are more than 97% identical to rRNA, scRNA, snoRNA, snRNA, and tRNA. The bowtie (1.1.2) software was used to identify known miRNAs by aligning clean data with the miRbase database. Furthermore, the MIREAP_v0.2 software (http://sourceforge.net/projects/mireap/) was used to predict novel miRNAs. The arguments and the thresholds used to identify miRNAs were specified in Appendix A.

### 2.6. Identification of Differentially Expressed Genes

Fragments per kilobase of transcript per million mapped reads (FPKM) normalization method was used to quantify the expression levels of lncRNAs and mRNAs. Tags per million (TPM) normalization method was used to quantify the expression levels of miRNAs. The edgeR [24] software was used to identify differentially expressed lncRNAs (DELs), mRNAs (DEMs), and miRNAs (DEMis). Genes with FDR < 0.05 and |log2FC| > 1 were considered as significantly differentially expressed (DEGs).

$$\mathrm{FPKM}\;=\;\frac{{C\;\times \;10}^{6}}{{\mathrm{NL}/10}^{3}}$$

Let FPKM be the expression level of transcript A, then C is the number of sequencing fragments aligned to transcript A, N is the total number of sequencing fragments aligned to the reference transcript, and L is the number of bases of transcript A.

$$\mathrm{TPM}\;=\;\frac{\mathrm{Actual}\;\mathrm{miRNA}\;\mathrm{counts}}{{\mathrm{Total}\;\mathrm{counts}\;\mathrm{of}\;\mathrm{clean}\;\mathrm{tags}\;\times \;10}^{6}}$$

### 2.7. Construction of LncRNAs-miRNAs-mRNAs ceRNA Regulatory Network

First, we predicted the DEMis targets within DELs and DEMs using MIREAP_v0.2, miRanda [25], and TargetScan [26] software. Secondly, we calculated the expression correlation between DEMis and its targets (lncRNA-miRNA or miRNA-mRNA) using the Spearman rank correlation coefficient (SCC) [27]. Pairs with SCC smaller than −0.7 were selected as candidate lncRNA-miRNA or miRNA-mRNA pairs. Thirdly, the expression correlation between DELs and DEMs was calculated using SCC, and pairs with SCC greater than 0.9 were selected as candidate ceRNA (lncRNA-mRNA) pairs. At last, the hypergeometric cumulative distribution function test in R software was used to identify final ceRNA pairs (*p*-value < 0.05). The lncRNA-miRNA-mRNA network was visualized using Cytoscape software v3.6.0 (http://www.cytoscape.org/) based on the target prediction, expression correlation, and hypergeometric cumulative distribution test results. The NetworkAnalyzer plug-in in Cytoscape software was used to calculate the connection degree. Nodes with connection degree greater than average degree of the whole ceRNA network were identified as highly-connected genes. The hypergeometric cumulative distribution function is as follow:$$P-\mathrm{value}\;=\;1\;-\;\overset{n-1}{\underset{i=0}{\sum }}\frac{\left(\begin{array}{c} M\\ i \end{array}\right)\left(\begin{array}{c} U-M\\ N-i \end{array}\right)}{\left(\begin{array}{c} U\\ N \end{array}\right)}$$
n is the number of miRNA that is shared by lncRNA and mRNA, U is the number of all miRNAs, M is the number of target miRNAs of lncRNAs, and N is the number of target miRNAs of mRNAs.

### 2.8. Functional Enrichment Analysis

To characterize the underlying function of the network, DEMs involved in the lncRNA-associated ceRNA network were performed to Gene Ontology (GO), and Kyoto Encyclopedia of Genes and Genomes (KEGG) enrichment analysis using clusterProfiler package [28] in R software based on the GO (http://www.geneontology.org/) and KO (http://www.kegg.jp/kegg/kegg1.html) databases. Gallus gallus was selected as the reference species for enrichment analysis. GO terms and KEGG pathways with *p*-value < 0.05 were considered as significantly enriched.

### 2.9. Quantitative Reverse Transcription Polymerase Chain Reaction

In the ceRNA network, three miRNAs were reported to be involved in adipocytes differentiation, including gga-miR-146a, gga-miR-135a, and gga-miR-128-1. gga-miR-6615 and gga-miR-135a were high-connected in the ceRNA network and high-expressed in chicken preadipocytes. These four miRNAs were involved in 12 ceRNA interactions such as TCONS_00026544-gga-miR-128-1-5p-RASD1, TCONS_00057272-gga-miR-146a-3p-FOXO6, TCONS_00057242-gga-miR-6615-3p-FOXO6, TCONS_00057242-gga-miR-6615-3p-ENSGALT00000043224, TCONS_00040913-gga-miR-6615-3p-ENSGALT00000043224, TCONS_00040913-gga-miR-6615-3p-ENSGALT00000050730, TCONS_00038747-gga-miR-6615-3p-ENSGALT00000050730, TCONS_00038748-gga-miR-6615-3p-ENSGALT00000050730, TCONS_00055280-gga-miR-135a-5p-JAM3, TCONS_00055280-gga-miR-135a-5p-GPR133, TCONS_00055280-gga-miR-135a-5p-CLDN1, TCONS_00055280-gga-miR-135a-5p-TMEM123. RT-qPCR was used to validate these ceRNA interactions. Each reaction was carried out for three biological and technical replicates. The primers for RT-qPCR are designed using the online Primer-Blast tool (Table 1). The 2^−ΔΔCt^ method was used to quantify the expression level of selected genes.

### 2.10. Statistical Analysis

The SPSS for Windows software (version 22, SPSS, Inc.) and Excel software (Microsoft Corp.) were used to analyze experimental data. The SPSS was used to test the normal distribution of the RT-qPCR data. The RT-qPCR data conformed to the normal distribution (Appendix A). Student t-test was used to analyze the difference in RT-qPCR data. All tests were performed at least in triplicate. The *p* < 0.05 value was considered to indicate a statistically significant difference.

## 3. Results 

### 3.1. Induction of Chicken Preadipocytes

The cells were induced to differentiation with differentiation medium supplemented with oleic acid. The oil red o staining and RT-qPCR were used to determine cell differentiation. We found that, after six days of induction, preadipocytes were fully differentiated and filled with large lipid droplets (Figure 1A,B). The relative expression level of adipocyte marker genes *FABP*4 and PPARG significantly increased (Figure 1C). 

### 3.2. Identification of Differentially Expressed lncRNAs, miRNAs, and mRNAs

A total of 3881 lncRNAs were identified in chicken preadipocytes (Appendix A). A correlation heat map was used to show the relationship between samples, which indicates that samples in the same group were highly correlated (Figure 2A). Differentially expressed genes between undifferentiated and differentiated preadipocytes were identified using edgeR software. Consequently, 235 DELs, 145 DEMis, and 660 DEMs were identified on the basis of RNA-Seq and small RNA-seq data (Appendix A). Heatmaps generated from the expression of DELs, DEMis, and DEMs were used to show the expression patterns of these genes between undifferentiated and differentiated preadipocytes (Figure 2B–D).

### 3.3. Construction of ceRNA Network Related to Preadipocytes Differentiation

By the microRNA response elements (MREs) prediction, we obtained 2262 interactions between 145 DEMis and 232 DELs, as well as 999 interactions between 139 DEMis and 192 DEMs. Based on SCC, 438 interactions between DEMis and DELs, 297 interactions between DEMis and DEMs, and 2883 interactions between DELs and DEMs were identified (Appendix A). Combined the MREs prediction and SCC results, DELs and DEMs that shared common DEMis were selected as candidate ceRNAs and tested using hypergeometric cumulative distribution function. In total, 251 ceRNAs (lncRNA-miRNA-mRNA) (*p* < 0.05) of 96 lncRNAs, 71 miRNAs, and 84 mRNAs were identified (Appendix A). The constructed lncRNA-miRNA-mRNA network was visualized using the Cytoscape software (Figure 3). Network analysis showed that the average degree of the whole network was 4.9.

### 3.4. Functional Enrichment Analyses

To further understand the underlying function of lncRNA-associated ceRNA network, GO and KEGG pathway analyses were performed on mRNAs involved in the ceRNA network. GO analyses showed that genes involved in the ceRNA network are associated with response to external stimulus, positive regulation of cell differentiation, and positive regulation of multicellular organismal biological processes (Figure 4). KEGG pathway analyses identified four significant enriched pathways, including pantothenate and CoA biosynthesis, mineral absorption, steroid hormone biosynthesis, and vascular smooth muscle contraction (Figure 5).

### 3.5. Identification and RT-qPCR Validation of Crucial Genes

In our study, miRNAs with TPM greater than 4.5 (more than two-thirds of the DEMis expression is lower than 4.5) were identified as highly expressed. Genes with connection degree greater than 4.9 were identified as highly connected. gga-miR-135a and gga-miR-6615 were high-connected in the network (degree > 4.9) and high-expressed in chicken preadipocytes (TPM > 4.5). gga-miR-146a, gga-miR-135a, and gga-miR-128-1 were reported to participate in the regulation of 3T3-L1 differentiation [29,30,31], suggesting their potential roles in chicken preadipocytes differentiation. Thus, ceRNA interactions containing these four miRNAs were identified as crucial ceRNA interactions. Seven lncRNAs and seven mRNAs interacting with these four miRNAs formed twelve ceRNA interactions (Figure 6, Table 2). Eight out of twelve ceRNA interactions were validated using RT-qPCR, demonstrating high consistency between RT-qPCR and RNA-seq results (Figure 7).

## 4. Discussion

In mammals, the molecular and cellular mechanisms underlying adipose tissue development have been well studied. The development of adipose tissue is a result of both increased number of new adipocytes (hyperplasia) and increased deposition of lipid in adipocytes (hypertrophy) [32]. However, adipose tissue development in chickens is still poorly understood. lncRNA was originally thought to be noise of RNA polymerase II transcription because it cannot encode a protein [33]. In recent years, with the development of high-throughput sequencing technology, more and more lncRNAs have been annotated. Increasing evidence has confirmed that lncRNAs are widely involved in the regulation of gene expression [34], cell proliferation [35], cell differentiation [36], cell apoptosis [37], and cancer development [38]. 

Recent studies suggest a significant number of lncRNAs play vital roles in regulating adipogenesis. For instance, lncRNA Lnc-U90926 attenuates 3T3-L1 adipocyte differentiation via inhibiting the transactivation of PPARγ2 or PPARγ [39]. PU.1 AS lncRNA, a novel AS lncRNA transcripted from the porcine PU.1 gene promotes adipogenesis through the formation of a sense-antisense RNA duplex with PU.1 mRNA [7]. However, the roles of lncRNAs in chicken adipogenesis remain elusive. To investigate the function of lncRNAs in chicken adipogenesis, we firstly isolated and cultured the preadipocytes from chicken abdominal fat tissue. Chicken preadipocytes were induced to differentiation in vitro using differentiation medium supplemented with oleic acid. Oil red O staining and RT-qPCR of adipocyte differentiation marker genes indicated that the chicken preadipocyte in vitro induction model was successfully constructed. We characterised the expression profiles of lncRNAs during the differentiation of chicken preadipocytes using Ribo-zero RNA-seq based on the in vitro induction model. A total of 3874 lncRNAs were identified, in which 235 lncRNAs were differentially expressed in the differentiation of chicken preadipocytes. Of the 235 DELs, 169 lncRNAs were up-regulated after differentiation, suggesting that most lncRNAs might play positive roles in the regulation of chicken preadipocytes differentiation. We also identified 145 DEMis and 660 DEMs between undifferentiated and differentiated preadipocytes. We found that top DEMs, such as FABP4 [40] and KLF5 [41] are involved in adipose tissue development of chicken. In the DEMis, gga-miR-128 [42] and gga-miR-223 [43] participate in regulating chicken preadipocytes differentiation. These differentially expressed genes were considered as potential key regulators in the differentiation of chicken preadipocytes.

Growing evidence suggests that lncRNAs can act as ceRNAs to indirectly regulate mRNAs by competitively binding miRNAs [10,15,44]. lncRNA-associated ceRNA networks in human adipocyte differentiation have been constructed [45,46], and lncRNAs regulating adipocyte differentiation as ceRNA in human [13], bovine [15], and mouse [47] have been identified. Nevertheless, the specific lncRNA-associated ceRNA network involved in the differentiation of chicken preadipocytes is yet largely unknown. In this study, we mainly focused on the construction of lncRNA-associated ceRNA network related to chicken preadipocytes differentiation. We firstly identified 235 DELs, 145 DEMis, and 660 DEMs between differentiated and undifferentiated preadipocytes. By target prediction and expression correlation analysis, 251 ceRNAs (lncRNA-miRNA-mRNA) (*p* < 0.05) of 96 DELs, 71 DEMis, and 84 DEMs were identified. We then constructed a ceRNA network based on these 251 ceRNA triples. To characterize the underlying function of the ceRNA network, we performed GO and KEGG pathway analysis on the 84 DEMs. GO analysis results revealed that DEMs were significantly enriched in positive regulation of cell differentiation biological process, suggesting the potential role of the constructed ceRNA network in cell differentiation. Pathway analysis also showed that DEMs were enriched in pathways related to preadipocyte differentiation, including pantothenate and CoA biosynthesis and steroid hormone biosynthesis.

To date, the function of lncRNA in chicken is poorly annotated. It is challenging to identify key lncRNA directly by its function. In our study, we identify crucial lncRNA-miRNA-mRNA interactions by miRNAs involved in adipogenesis and miRNAs with high connection degree and expression. Studies showed that miR-128 promoted adipogenic differentiation of human mesenchymal stem cells by suppression of VEGF pathway [29]. The miR-146a can promote BM-MSC to differentiate into adipocytes [30]. miR-135a-5p inhibits 3T3-L1 adipogenesis through activation of canonical Wnt/β-catenin signaling [31]. gga-miR-6615-3p were high-connected in the network and high-expressed in chicken preadipocytes. These four miRNAs may play crucial roles in the differentiation of chicken preadipocytes. Finally, we identified 12 vital lncRNA-miRNA-mRNA interactions containing the above four miRNAs from the lncRNA-associated ceRNA network. Eight out of the twelve interactions were validated by RT-qPCR: TCONS_00026544-gga-miR-128-1-5p-RASD1, TCONS_00057272-gga-miR-146a-3p-FOXO6, TCONS_00057242-gga-miR-6615-3p-FOXO6, TCONS_00057242-gga-miR-6615-3p-ENSGALT00000043224, TCONS_00055280-gga-miR-135a-5p-JAM3, TCONS_00055280-gga-miR-135a-5p-GPR133, TCONS_00055280-gga-miR-135a-5p-CLDN1, TCONS_00055280-gga-miR-135a-5p-TMEM123. The expression of gga-miR-128-1-5p and gga-miR-135a-5p decreased during the differentiation of chicken preadipocytes. It suggests that lncRNA TCONS_00026544 and TCONS_00055280 may promote adipogenic differentiation of chicken preadipocytes by sponging gga-miR-128-1-5p and gga-miR-135a-5p, thereby up-regulating the expression of target genes, RASD1, JAM3, GPR133, CLDN1, and TMEM123. Instead, gga-miR-146a-3p and gga-miR-6615-3p were up-regulated in differentiated cells, demonstrating their positive role in regulating the adipogenesis of chicken. lncRNA TCONS_00057272 and TCONS_00057242 may sponge gga-miR-146a-3p and gga-miR-6615-3p as ceRNA, increase the expression of their target genes FOXO6 and ENSGALT00000043224, thereby inhibit the differentiation of chicken preadipocytes. However, the contribution of the crucial lncRNA-miRNA-mRNA interaction identified in our study is still not certain. Further studies should be performed to address these issues.

## 5. Conclusions

In summary, we systematically analyzed the differentially expressed lncRNAs, miRNAs, and mRNAs between differentiated and undifferentiated preadipocytes using RNA-seq and miRNA-seq. We then constructed the lncRNA-associated ceRNA network following the ceRNA hypothesis. Eight crucial ceRNA interactions were identified based on the interacted miRNAs function and RT-qPCR. TCONS_00026544-gga-miR-128-1-5p-RASD1, TCONS_00055280-gga-miR-135a-5p-JAM3, TCONS_00055280-gga-miR-135a-5p-GPR133, TCONS_00055280-gga-miR-135a-5p-CLDN1, and TCONS_00055280-gga-miR-135a-5p-TMEM123 may promote adipogenic differentiation of chicken preadipocytes while TCONS_00057272-gga-miR-146a-3p-FOXO6, TCONS_00057242-gga-miR-6615-3p-FOXO6, and TCONS_00057242-gga-miR-6615-3p-ENSGALT00000043224 have the opposite effects. Our study constructs a ceRNA regulatory network and identifies eight crucial ceRNA interactions related to chicken preadipocytes differentiation, which will lead to a better understanding of chicken fat deposition. However, the ceRNA network and crucial ceRNA interactions is results of prediction based on gene expression and target prediction, and their function in the differentiation of chicken preadipocytes still needs to be verified by experiments.

## Figures and Tables

**Figure 1 genes-10-00795-f001:**
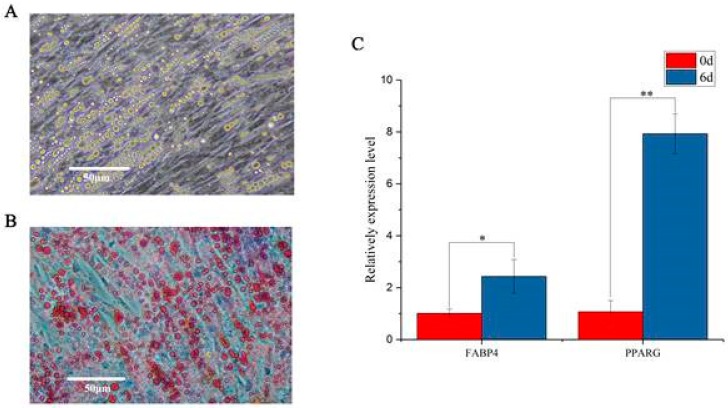
Differentiation of chicken preadipocytes. (**A**) Differentiated preadipocytes before oil red o staining. (**B**) Differentiated preadipocytes stained using Oil red O. (**C**) The relative expression level of adipocyte marker genes before and after differentiation. The relative expression level of FABP4 and PPARG genes significantly increased post-differentiation, indicating that the chicken preadipocytes were successfully induced. *denotes *p* < 0.05, **denotes *p* < 0.01.

**Figure 2 genes-10-00795-f002:**
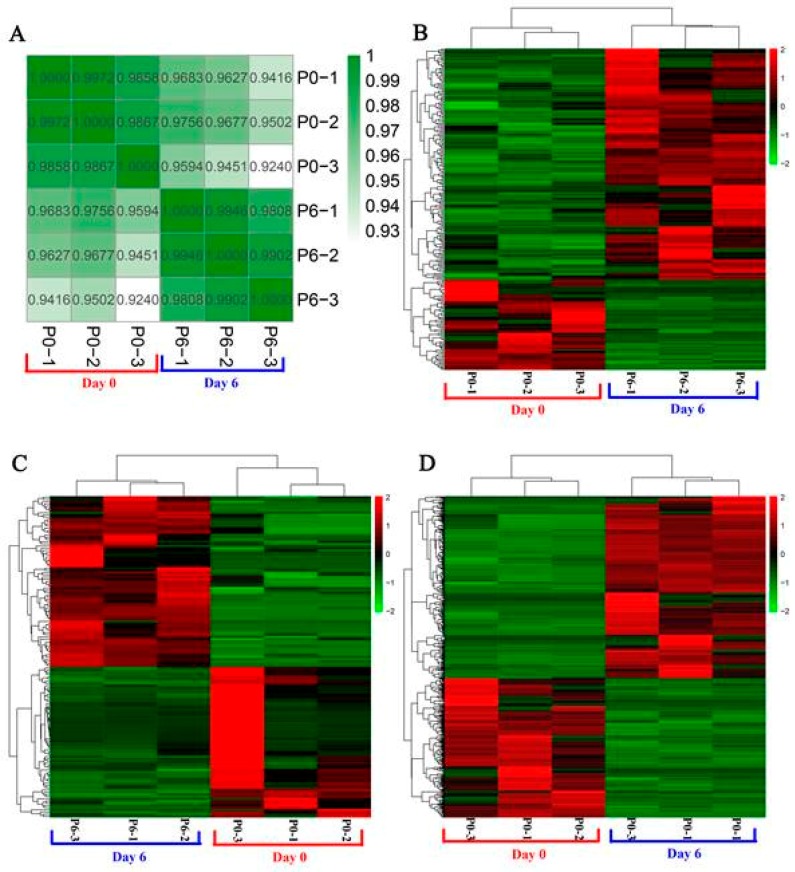
Heatmaps of sample correlation and differentially expressed genes. (**A**) Sample correlation heatmap. Dark green indicates a high correlation. (**B**) Heatmap for differentially expressed long noncoding RNAs (lncRNAs). (**C**) Heatmap for differentially expressed microRNAs (miRNAs). (**D**) Heatmap for differentially expressed mRNAs. Red indicates highly expressed genes, and green indicates lowly expressed genes.

**Figure 3 genes-10-00795-f003:**
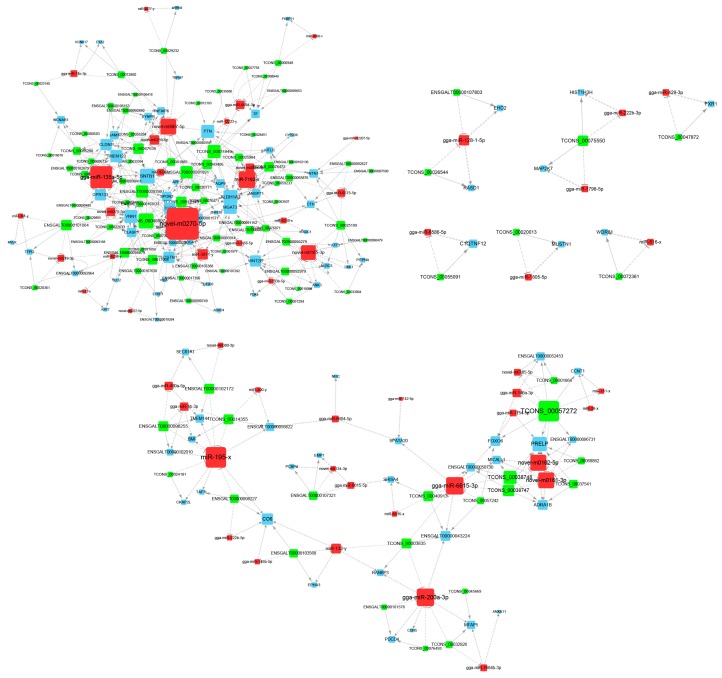
Visualization of the lncRNA-miRNA-mRNA network. Green nodes indicate lncRNAs, red nodes indicate miRNAs, and blue nodes indicate mRNAs, respectively. Node size is proportional to the connection degree. Connection degree shows the number of connected nodes with the individual node. The higher the degree of a node, the more important the node in the network. Hub genes are the nodes with higher degree i.e., nodes with more connections (degree > 4.9).

**Figure 4 genes-10-00795-f004:**
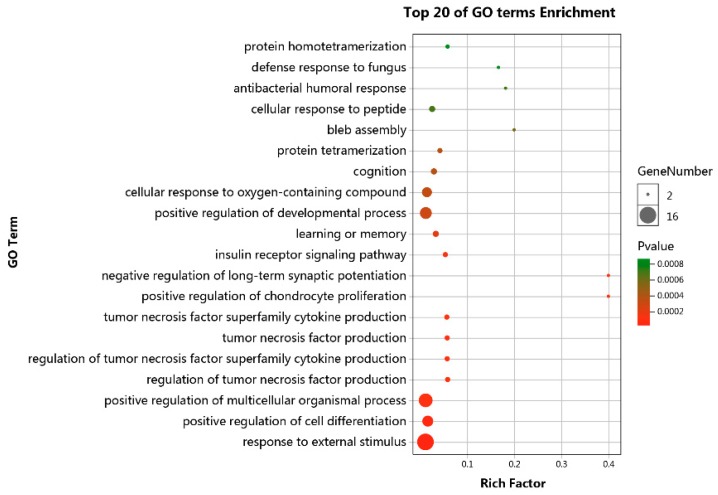
GO analyses of mRNAs involved in the lncRNA-related competing endogenous RNA (ceRNA) network.

**Figure 5 genes-10-00795-f005:**
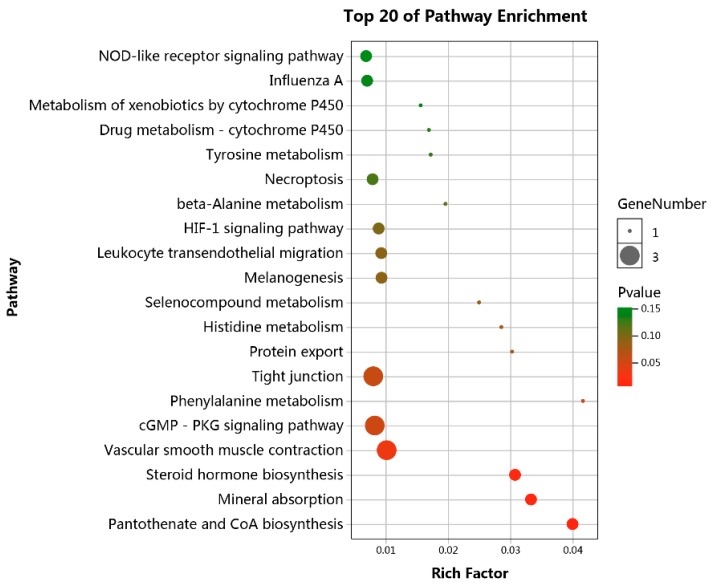
KEGG pathway analyses of mRNAs involved in the lncRNA-related ceRNA network.

**Figure 6 genes-10-00795-f006:**
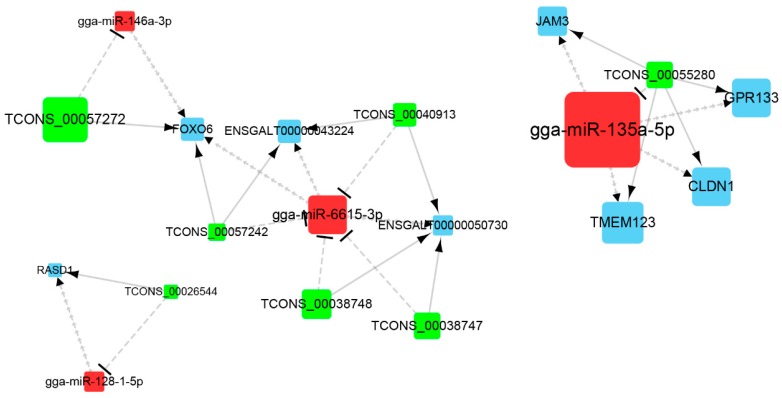
Visualization of crucial genes in the ceRNA network. Green, red, and blue nodes indicate lncRNAs, miRNAs, and mRNAs, respectively.

**Figure 7 genes-10-00795-f007:**
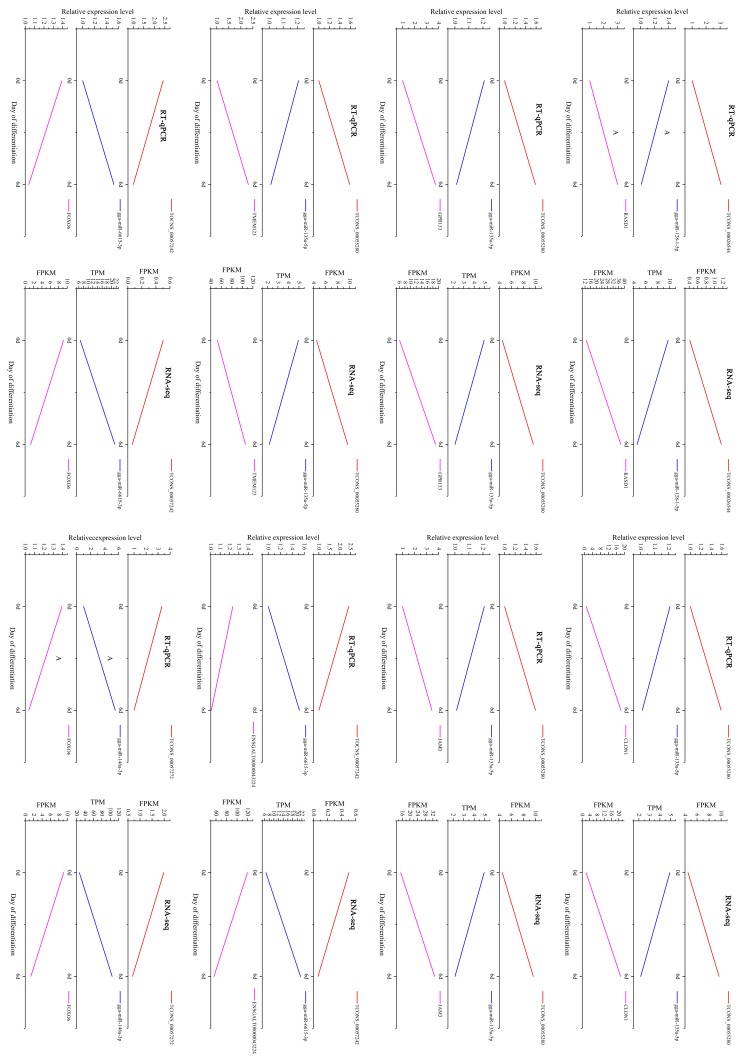
Validation of crucial ceRNA interactions using quantitative reverse transcription polymerase chain reaction (RT-qPCR). Red, blue, and pink lines indicate lncRNA, miRNA, and mRNA, respectively. Eight out of twelve ceRNA interactions were validated using RT-qPCR, including TCONS_00026544-gga-miR-128-1-5p-RASD1, TCONS_00057272-gga-miR-146a-3p-FOXO6, TCONS_00057242-gga-miR-6615-3p-FOXO6, TCONS_00057242-gga-miR-6615-3p-ENSGALT00000043224, TCONS_00055280-gga-miR-135a-5p-JAM3, TCONS_00055280-gga-miR-135a-5p-GPR133, TCONS_00055280-gga-miR-135a-5p-CLDN1, TCONS_00055280-gga-miR-135a-5p-TMEM123.

**Table 1 genes-10-00795-t001:** Primers for RT-qPCR.

Gene	Primer sequences
TOCNS_00055280	GCAGAACGGATGGTGCCTTTGT
	GGCTGTCCTGGCTGAAGATGGA
TOCNS_00057242	CCCTTCTCCGCTCACAGTCCTT
	CGCTTCGGCTGTAAACGTCCAT
TOCNS_00057272	GGCTACCCGTCTCCTCCAAGAT
	GTGCCTCGTTACGCCTGATTGA
TOCNS_00026544	ACGGAGATGCTGCGGTTATGC
	AACAAACCCAACCCGTTCCCAG
ENSGALT00000043224	TCCAAGAAGGCGGTCACCAAGA
	GCACCTGCTTCAGCACCTTGT
FOXO6	CAGCAGACCTGGACCTGGACAT
	CGCCGAGTCGAAGTTGAAGTCC
RASD1	CCTCGTGTTCAGCCTGGACAAC
	CGGCACCTCGATGTTCTCCTTG
GPR133	GTGAGCACCATCCGCAACCAA
	AAGGCACCGTTCCAGGACTGAA
JAM3	GGAAGTCCTCCTCGCAGCAGTT
	AACAAGCCAGGTGCCCACTCT
CLDN1	GCTGATTGCTTCCAACCAGGCT
	GCACACGGCTCTCCTTGTCTAC
TMEM123	CGTACCACTCGAGAAGAGGC
	CAGCTGTGACAGGATGGGTT
gga-miR-6615-3p	GTCGTATCCAGTGCAGGGTCCGAGGTATTCGCACTGGATACGACTGTGGA
	CGCGTGGCACTGATGTGTTC
gga-miR-146a-3p	GTCGTATCCAGTGCAGGGTCCGAGGTATTCGCACTGGATACGACCTGAAG
	CGACCCATGGGGCTCAGTT
gga-miR-128-1-5p	GTCGTATCCAGTGCAGGGTCCGAGGTATTCGCACTGGATACGACTCTCAG
	GCGGGGCCGTAACACTGT
gga-miR-135a-5p	GTCGTATCCAGTGCAGGGTCCGAGGTATTCGCACTGGATACGACTCACAT
	CGCGTATGGCTTTTTATTCCT

**Table 2 genes-10-00795-t002:** The crucial ceRNA Interactions.

lncRNA	miRNA	mRNA
TCONS_00026544	gga-miR-128-1-5p	RASD1
TCONS_00057272	gga-miR-146a-3p	FOXO6
TCONS_00057242	gga-miR-6615-3p	FOXO6
TCONS_00057242	gga-miR-6615-3p	ENSGALT00000043224
TCONS_00040913	gga-miR-6615-3p	ENSGALT00000043224
TCONS_00040913	gga-miR-6615-3p	ENSGALT00000050730
TCONS_00038747	gga-miR-6615-3p	ENSGALT00000050730
TCONS_00038748	gga-miR-6615-3p	ENSGALT00000050730
TCONS_00055280	gga-miR-135a-5p	JAM3
TCONS_00055280	gga-miR-135a-5p	GPR133
TCONS_00055280	gga-miR-135a-5p	CLDN1
TCONS_00055280	gga-miR-135a-5p	TMEM123

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
