# Peer review of "Identification of Long Non-Coding RNA-Associated Competing Endogenous RNA Network in the Differentiation of Chicken Preadipocytes"

_genes, 2019, doi:10.3390/genes10100795_

Round 1

Reviewer 1 Report

Review Genes 593445

The authors performed the identification of interaction networks among lncRNAs-miRNAs-mRNAs differentially expressed between differentiated and undifferentiated preadipocytes. The paper has relevant results that might contribute to the knowledge about the biological processes associated with fat deposition in chicken. However,  some details must be clarified. See below my comments.

Major comments:

In general, the manuscript is well-written and the results are relevant. My main concerns are related to the methods and discussion sections. 

Methods:

Line 79: The authors must specify from with cells and from which cell concentration the RNA was extracted. 

Line 84: The 12 constructed libraries must be presented in a clear way. I suggest the authors add the number os libraries constructed for each group between parentheses. 

Lines 85-97: The criteria for the lncRNAs and miRNA detection must be better specified. Which were the arguments and the thresholds used to identify each of the RNA classes in the different software?

Lines 199-100: FPKM and TPM are not algorithms. The authors used an algorithm to estimate FPKM and TPM. However, these values are a metric for gene expression. 

Line 103: Did the authors applied any multiple-testing correction (FDR, Bonferroni, etc.) for the test-statistics before identifying the DE mRNAs, miRNAs and lncRNAs?

Lines 108-110: The authors must provide a reference or a justification for why the SCC thresholds applied were chosen. 

Lines 110-111: Which software was used to perform the hypergeometric cumulative test?

Lines 112-113: Was the network built based on the correlation values? This must be clearly specified.

Lines 115-117: Which software was used to perform the enrichment analysis? Additionally, which were the thresholds defined to classify a term as enriched? At least, which was the species used during the enrichment analysis? Did the authors perform the enrichment using the Gallus gallus ad reference? If yes, did the authors considered to run the enrichment using an evolutionarily related species with a better GO and KEGG annotation?

Lines 119-120: Inform the selected interactions and why these were selected.

Lines 124-127: Did the data showed a normal distribution? This must be tested to choose the proper statistical test.

The last comment about the methods section is regarding the determination of differentiated and undifferentiated preadipocytes. The first results showed in the “Results” section are about the cytological and “marker genes expression” to determine the differentiation status. However, there any mention of this procedure in the material and methods section. This must be very well specified. 

Discussion:

Lines 397-419: These lines are not discussed. This is a summary of the results. The authors must use this section to provide more scientifical evidence that supports their results. For example, the authors might expand the discussion about the DE miRNAs, lncRNAs and mRNAs. There is no discussion about the association of the target genes and fat deposition (or adipogenesis). The discussion must be improved substantially.

Minor comments:

Lines 287-288: How the authors identified that these miRNAs are known to be involved with adipocytes differentiation? Literature review? If yes, provide the references.

Lines 291-292: Did the authors tested the list of twelve ceRNA for GO and KEGG pathways enrichment?

Lines 455-457: “Our results not only provide novel insights into ceRNA roles of lncRNAs in chicken preadipocytes differentiation and but also contribute to a better understanding of chicken fat deposition.”. This sentence is repeated exactly in the same way in the abstract, introduction and conclusions. This must be avoided. 

Figure 1: Probably there is a typo close to (C).

Figure 2: The Figure should clearly show which animals are from each group in the dendrogram. The dendrogram branches could be plotted with different colors or a bar with different grouping colors could be added to the plot. 

Figures 4 and 5: The legend of the figure must be more descriptive.

Figure 7: Describe the colors in the same order as the figures.

Reviewer 2 Report

1. The introduction can be strengthened by a short explanation of ceRNA activity.

2. It seems to me that the aim and the significance of this study are not well defined in the introduction. The authors started with the problem of excess fat deposition, but how those ceRNA networks contribute to fat deposition was not discussed. It's confusing what the aim was, understanding adipocytes differentiation or understanding fat deposition. 

3. Line 90 and line 97, were any criteria applied to filter lncRNA and mRNA prediction results?

4. Line 103, I would suggest using adjusted P-value (or FDR) of 0.05 to claim statistical significance of differentially expressed genes. 

5. Line 115-117, what software/tool was used for GO term and KEGG analysis?

6. Line 236, what is connection degree? It should be explained in the manuscript.

7. Line 289 and Table 2, were there any quantitative criteria to define highly-connected, highly-expressed and crucial?

8. Line 403, change "Using" to "using"

9. Line 416, it seems not logically sound to claim that ceRNA interactions are involved in certain biological processes or pathways when the DEMs are enriched in those biological processes or pathways. The claim can be true if the two following assumptions are met: a, the predicted interaction is valid, and b, this ceRNA interaction does happen in adipocytes differentiation. But from this study, whether or not those two assumptions are met could not be proved. 

10. As for the validation of ceRNA network, RT-qPCR can only confirm the relationship of their expression pattern. Are there any other methods to validate ceRNA interaction network?

Round 2

Reviewer 2 Report

It looks like clusterProfiler was used for GO and KEGG pathway analysis. If so, please add the citation. In my opinion, the newly-added sentences for the caption of Figure 4 and 5 are not necessary. I suggest deleting them or rephrasing in a more scientific way. The explanation of Figure 6 can be moved to results or methods instead. Line 554: delete “first” I would suggest adding the limitation of this study in the discussion and conclusion.
